# Use of Three-Dimensional Cell Culture Models in Drug Assays for Anti-Cancer Agents in Oral Cancer: Protocol for a Scoping Review

**DOI:** 10.3390/jpm13111618

**Published:** 2023-11-17

**Authors:** Everton Freitas de Morais, Leonardo de Oliveira Siquara da Rocha, John Lenon de Souza Santos, Raíza Dias Freitas, Bruno Solano de Freitas Souza, Ricardo D. Coletta, Clarissa A. Gurgel Rocha

**Affiliations:** 1Department of Oral Diagnosis, School of Dentistry, University of Campinas (UNICAMP), Piracicaba 13414-903, SP, Brazil; evertonf@unicamp.br (E.F.d.M.); coletta@unicamp.br (R.D.C.); 2Gonçalo Moniz Institute, Oswaldo Cruz Foundation (IGM-FIOCRUZ/BA), Salvador 40296-710, BA, Brazil; leonardo.oliveira@ufba.br (L.d.O.S.d.R.); john.santos@fiocruz.br (J.L.d.S.S.); bruno.solano@fiocruz.br (B.S.d.F.S.); 3Department of Pathology and Forensic Medicine, School of Medicine, Federal University of Bahia, Salvador 40110-100, BA, Brazil; 4Department of Social and Pediatric Dentistry, School of Dentistry, Federal University of Bahia, Salvador 40110-150, BA, Brazil; raiza.dias@ufba.br; 5D’Or Institute for Research and Education (IDOR), Salvador 41253-190, BA, Brazil; 6Graduate Program in Oral Biology, School of Dentistry, University of Campinas, Piracicaba 13414-903, SP, Brazil; 7Department of Propaedeutics, School of Dentistry, Federal University of Bahia, Salvador 40110-150, BA, Brazil

**Keywords:** mouth neoplasms, cell culture techniques, three-dimensional, organoids, drug-screening assays, antitumor, pharmacology

## Abstract

Advances in the development of pharmacological treatment in oral cancer require tumor models capable of simulating the complex biology of the tumor microenvironment. The spread of three-dimensional models has changed the scenery of in vitro cell culture techniques, contributing to translational oncology. Still, the full extent of their application in preclinical drug trials is yet to be understood. Therefore, the present scoping review protocol was established to screen the literature on using three-dimensional cell culture models in drug-testing assays in the context of oral cancer. This scoping review will be conducted based on the guidelines established by the Preferred Reporting Items for Systematic Reviews and Meta-Analyses Extension for Scoping Review guidelines (PRISMA-ScR). We will search the PubMed/Medline, Web of Science, Scopus, and Embase databases, as well as the gray literature, including peer-reviewed research articles involving 3D models applied to drug-assessment assays in oral cancer published from 1 March 2013 until 1 March 2023. Data will be charted, and findings will be described according to the predetermined questions of interest. We will present these findings in a narrative manner.

## 1. Introduction

A persistent challenge in oncology research is managing the complex biology behind tumor behavior and progression. This is true for aggressive diseases such as oral cancer, which remains among the most frequent and debilitating types of cancers worldwide [1,2]. Its most common subtype, oral squamous cell carcinoma (OSCC), usually requires extensive surgical treatment, significantly impacting patient outcomes [3,4]. Despite existing treatments, the average 5-year survival rate remains around 50%, especially among advanced cases [5]. Developing therapies against oral cancer cases is challenged by the tumor’s genetic heterogeneity, rapidly evolving phenotypes, and immune- and treatment-defense strategies [6].

The focus of preclinical research in oral cancer has shifted from studying only the tumor cell to studying the tumor microenvironment (TME), whose heterogeneous cell population and transformable extracellular matrix (ECM) define the way tumors behave and respond to treatment [7]. The non-tumoral cells within the OSCC TME not only may represent up to 90% of the tumor mass [8], but also hold an important place in shaping tumor growth and dissemination [9]. Cancer-associated fibroblasts, the main stromal cell type in OSCC, modulate the TME, influencing tumor progression from cell proliferation and metastasis to drug resistance [10]. Aside from cellular populations, the extracellular matrix can also modulate tumor behavior through characteristics such as tumor stiffness [10,11], metabolism [10,12,13], and secretion of growth factors promoting tumor initiation and invasion [14,15]. The mirroring of the dynamics and composition of the TME in preclinical studies requires models that replicate cellular spatial assembly and behavior.

Cell culture systems have significantly impacted the field of biology by reducing the necessity for laboratory animal use while advancing research, pharmaceutical discoveries, and the evolution of medicine [16]. Initially, cells were cultivated in a two-dimensional (2D) format, adhering to polystyrene utensils or flat surfaces [17]. However, researchers soon began culturing cells in three-dimensional (3D) environments with attachment proteins within a synthesized ECM [18,19]. The traditional 2D in vitro cell culture system involves cell growth as a monolayer on a flat surface [17], a practice that dates to the early 1900s. This method has been historically utilized in research, especially for co-culturing cellular heterogeneity and evaluating the biological performance of bioactive molecules in oncological research [17]. Nonetheless, this culture system has several limitations, as it fails to accurately replicate functional conditions and the natural microenvironment, including structure, physiology, biological signals of living tissues, as well as cell–cell and cell–matrix interactions [18]. In this context, three-dimensional culture systems have become popular due to their ability to mimic tissue-like structures more effectively than two-dimensional cultures [16]. Several 3D methods have been developed and evaluated for multiple purposes, such as disease modeling, drug testing, and identifying new therapeutic targets [16,17].

Preclinical drug trials in cancer are reliant on models that allow for the pharmacological testing of tumor cell lines, such as adherent cell culture models, which are still a common resource for researchers worldwide for their technical simplicity, financial feasibility, and reproducibility inside a laboratory [2]. Nevertheless, 2D cultures may not sufficiently mimic the physiological conditions of cells, as they do not reflect the complex architecture and the three-dimensional interactions that occur among cells in vivo. Differently from what can be simulated in 2D laboratory models, pharmacological distribution and response inside a tumor are influenced by the organization of TME cells and their interactions with the ECM. Therefore, deceptive data from 2D cell culture models often lead to the irrelevant prediction of drug efficacy and toxicity, failing drug validation and approval processes [16,17,18].

The application of 3D culture models in vitro to reconstitute essential aspects of the TME, such as cell heterogeneity [19,20], nutrient distribution [21,22], and oxygen gradients [23,24], is now widely discussed as a proper model for preliminary drug research in oncology. New tools are being developed to enable the accurate modeling of tumor responses to pharmacological testing [2,4] by incorporating non-cellular components, such as ECM proteins, and developing heterotypic models containing stromal cells, such as fibroblasts, endothelial cells, or even immune cells [6]. The abundance of distinct 3D culture methods and protocols confuses the criteria for their application and eligibility, considering the desired assays and objectives. Nonetheless, 3D models such as spheroid and organoid cultures promise superior results over standard monolayer techniques and are positioned as a helpful tool in pharmacological testing and molecular and cellular studies of cancer progression. Their usage potentially increases the accuracy of drug selection and the advancement of clinical trials, meaning a quicker development of personalized medicine [3].

This current study aims to evaluate how 3D in vitro methods have been employed in drug-testing assays in oral cancer, highlighting how available 3D technologies translate the reality of oral cancer’s susceptibility to anticancer drugs. A scoping review will be performed to map the current state of evidence and identify the lacunae in the literature related to the role of 3D culture methods applied to oral cancer and preclinical trials of anti-cancer drugs.

## 2. Materials and Methods

### 2.1. Research Questions

Considering this review’s aim, we intend to answer the following research questions:What protocols and techniques are recommended for applying three-dimensional culture methods in preclinical trials of anti-cancer drugs targeting oral cancer?Which 3D culture models have been predominantly utilized in preclinical drug trials for oral cancer treatment, and what are their primary applications?What are the documented advantages and challenges of using 3D culture models in preclinical drug trials for oral cancer as compared to traditional 2D models?

### 2.2. Study Design

This scoping review protocol was formulated based on the guidelines set by Arksey and O’Malley [25] and aligns with the Joanna Briggs Institute’s recommendations for constructing scoping reviews [26]. We used the population–concept–context (PCC) mnemonic to define the research question and shape the eligibility criteria and the literature search, as shown in Table 1. The reporting of this scoping review was based on the title, introduction, and methods section of the PRISMA Extension for Scoping Reviews (Appendix A) [27].

### 2.3. Search Strategy

The search strategy for PubMed/Medline was formulated (Table 2) and then customized to the databases Embase, Scopus, and Web of Science (Appendix A). Additionally, we will explore the gray literature through Google Scholar. We designed the search strategy by integrating Medical Subject Headings (MeSH) terms, their entry terms, and other relevant keywords using the Boolean operators “AND”, “OR”, and “NOT”. The search strategy was based on three concept clusters: (1) oral cancer, (2) three-dimensional cell culture models, and (3) drug-screening assays.

The methods of this study are summarized in Figure 1.

Searches will be performed, and results will be exported in Comma-Separated Values (CSV) format to Microsoft Excel (version 2302). Deduplication will be performed manually by two independent reviewers, who will proceed with the application of the inclusion and exclusion criteria.

### 2.4. Study Selection

#### 2.4.1. Inclusion Criteria

Studies that have been published in the last ten years were initially included before proceeding with exclusion criteria. All articles found were included regardless of language or journal source.

#### 2.4.2. Exclusion Criteria

Studies were excluded if they met any of the following criteria:Reviews of any sort, book chapters, author’s opinion/comments, editorials, meeting abstracts, conference abstracts and study protocols, and articles without available full text;Studies not regarding oral cancer;Studies that did not employ in vitro three-dimensional methods of any sort;Studies that did not perform preclinical drug trials or evaluations.

The screening of studies was conducted by two authors independently. Inter-rater agreement was assessed through Cohen’s κ at the abstract review stage. Discrepancies were resolved by discussion or a third reviewer if they remained. Reviewers screened the search results initially based on the studies’ titles, proceeding to assess their abstracts and full text when necessary.

### 2.5. Data Extraction and Charting

Data extraction was performed by two reviewers according to predetermined extraction items. Data from eligible studies was tabulated in a Microsoft Excel sheet under headings compatible with the items defined in this protocol. After extraction, data was discussed among the authors. Since no standard checklist exists for in vitro studies, we created our items of assessment based on the most essential components of in vitro studies and relevant information to the question of our study. The items were answered by yes or no, followed by explanatory answers when necessary. No studies were excluded based on the quality assessment.

The following items were assessed during data extraction:What subtype or subtypes of oral cancer were evaluated (e.g., cell line, cell bank, origin, topographical location on the mouth)?What were the cell culture conditions (culture medium, additives, atmosphere, temperature)?What was the three-dimensional model or models used (e.g., spheroids, organoids, organ-on-a-chip) and their specifications (e.g., protocol steps, use of scaffolds, patient-derived explants)?What were the evaluated compounds and their relevant information (e.g., purchase, concentration)? Were they antineoplastic agents?What was the treatment protocol (e.g., application scheme, duration, association with other treatments)?What were the control group(s)/conditions?What assays were performed to evaluate compound effects (e.g., cell death, cytotoxicity, invasion, migration)? What were their results?What statistical analysis was performed? How many samples (e.g., number of spheroids or organoids) were used? How many assay repetitions were made?What are the authors’ conclusions?Overall, does the study mention all data accurately?

Quality of evidence was evaluated by the GRADE method [28] adapted for in vitro studies [29]. Two independent reviewers categorized the included articles as “high”, “moderate”, “low”, and “very low” quality. Discrepancies were discussed among the authors to reach a final decision. Based on these evaluations, articles were classified accordingly [30].

### 2.6. Analysis Plan

#### 2.6.1. Data Analysis Approach

We applied qualitative analysis of the extracted data to understand how 3D culture methods have been employed in preclinical trials of anti-neoplastic agents in oral cancer.

#### 2.6.2. Data Summary

We intend to present our results in narrative form, including tables containing the relevant extracted information. Findings will be described according to the review questions objectively, and the results section may undergo further adjustments after the results are reviewed. As a guideline, the PRISMA-ScR checklist was used in this review.

Any further changes to the study protocol were made as necessary and reported accordingly.

## 3. Discussion

Oral cancer, the sixth leading cause of cancer-related deaths globally [2], is often associated with early and extensive lymph node metastases, contributing to its classification among malignancies with notably low survival rates. Despite advancements in diagnostic and therapeutic strategies for oral cancer over recent decades, the five-year survival rate remains below the desired levels [1]. Currently, early-stage non-metastatic oral cancer (stages I and II) can often be effectively treated with surgery alone. However, for advanced oral cancer (stages III and IV), conventional surgical intervention and external radiotherapy must be supplemented with supportive treatment involving a combination of chemotherapeutic agents [1]. Novel drug establishment faces many challenges in current oncology research. The cost of securing new medication has more than doubled in ten years [31], and new compounds may take 10–15 years to become available to the public [30]. Aside from being time-consuming and costly, drug development in oncology maintains a low success rate, with under 5% of tested drugs reaching the pharmacy shelves and the average patient [30,32,33]. One of the challenges behind this process is the difficulty in the translation of research from the laboratory to clinical practice. Traditional laboratory models, such as 2D cell culture or xenograft animal models, do not adequately capture the dynamics of the TME [17,34,35]. In the same way that characterizing the tumor microenvironment is a significant step in identifying prognosis-related and targetable markers [9], reproducing such characteristics in laboratory pharmacological research is crucial for therapy development. Developing and disseminating three-dimensional models is an important step in bridging this gap. Nevertheless, considering the diversity of methods, technologies, and applications [36,37,38], an appraisal of the role of these models in preclinical drug trials is needed. 

To meet the increasing demand for a single technology that adequately caters to 3D cell culture needs, various methods have been developed. The utilization of 3D culture models allows for the examination of morphological and cellular arrangements influenced by ECM interactions, crucially altered in oncogenic transformation. Consequently, these 3D in vitro tumor models play a vital role in studying cancer growth mechanisms and metastasis [39]. Employing appropriate 3D culture approaches offers a more physiologically relevant method to analyze gene expression and cell phenotype outside of their natural environment [38]. The engineering of 3D cultures is guided by distinct fundamental principles: the cell nature (from its selection and isolation from the original tissue to considerations regarding specific cell lines), artificial 3D microenvironment (which stimulates or allows for ECM production), biomaterial-based scaffolding (which may be natural, synthetic, or rigid), signaling molecules (proteins and growth factors), and bioreactors (for cell culture) that sustain a biologically active cellular milieu [38,40,41]. Hence, these parameters necessitate thorough evaluation before selecting the most pertinent technique and model. Culture systems can be scaffold-based, relying on natural or artificial solid scaffolds, or scaffold-free, such as spheroids (non-scaffold based) [38].

In the 3D cell culture environment, a significant increase in the resistance of neoplastic cells to treatment with chemotherapy drugs is noted. Heightened resistance to cytotoxic drugs within 3D culture models can be attributed to various factors. These encompass an elevated expression of cancer stem cell-associated proteins [40,41,42,43], the upregulation of numerous drug resistance genes and microRNAs [42], overexpression of multi-drug resistance proteins in cultured cells, hindered drug penetration in multicellular 3D models, and remodeling of the ECM [42,43]. Moreover, the exposure of cells in a 3D culture to varying levels of essential compounds in a gradient may elucidate the divergent responses observed across different models [43]. The interactions of cells among themselves and with the ECM also significantly influence the cellular response to drugs. Additionally, the structural composition and density of 3D spheroids, which is developed distinctly among different cancer cell lines, can impact their response to drugs. Cells forming dense spheroids exhibit heightened resistance to paclitaxel and doxorubicin compared to cells in 2D culture. Conversely, cells forming loose 3D spheroids demonstrate a drug response comparable to 2D-cultured cells [31,35]. Matrix stiffness is another critical factor affecting drug resistance in 3D culture models and plays a substantial role in determining cellular responses to anti-cancer agents. A study by Ki et al. demonstrated that the immobilization of an EGFR inhibitor (NYQQNC) on PANC-1 cells led to reduced cell viability in stiff hydrogels but not in hydrogels with low stiffness [44].

Oral cancer is distinguished by a heightened presence of immune cells infiltrating the tumor, defining it as a highly immunogenic tumor [45]. Within the intricate tumor microenvironment, which encompasses the ECM, diverse stromal cells and immune cells coordinate a dynamic interplay with the tumor cells. Notable interacting cell types include tumor-associated macrophages (TAMs), regulatory T cells (Tregs), cancer-associated fibroblasts (CAFs), and endothelial cells [46]. Moreover, the innate immune system is present in the TME as macrophages, dendritic cells (DCs), neutrophils, myeloid-derived suppressor cells (MDSCs), natural killer cells (NKs), and innate lymphoid cells. The adaptive immune response is represented by T cells and B cells [47]. The communication and interplay among these cells, as well as with the ECM and tumor cells, play a pivotal role in propelling tumor progression [46,47] and must be considered when developing a three-dimensional culture model.

Diverse 3D monoculture and double/triple co-culture systems exhibit distinct responses of cancer cells to anti-cancer drugs based on the composition of the ECM, cellular interactions, and soluble factors secreted by the cells [40]. In a separate study, Huh-7 cells cultured in a 2D model demonstrated greater sensitivity to sorafenib compared to those in a 3D monoculture. Additionally, cells in the 3D culture exhibited higher sensitivity to sorafenib than Huh-7 cells co-cultured with hepatic stellate cells [47]. The tumor microenvironment further influences drug response in triple culture scenarios. Non-small cell lung cancer (NSCLC) cells (NCI-H157) displayed increased resistance to paclitaxel when co-cultured with CAFs and monocytes compared to monocultured cells [48]. However, interaction with neighboring cells in the tumor microenvironment does not consistently enhance drug resistance. The response of NCI-H157 to cisplatin remained unchanged in both monoculture and triple culture conditions with CAFs and monocytes [49]. Consequently, understanding the drug responses of patient-derived cells cultured in multicellular tumor models can provide insights into in vivo drug responses, making such models pivotal in the realm of precision medicine.

Tumor organoids possess the potential to serve as an in vitro representation of a patient’s tumor, replicating its molecular and phenotypic diversity. This characteristic makes tumor organoids valuable for predicting how individual patients might respond to specific treatments in the realm of personalized medicine. Pasch et al. [49] successfully developed tumor organoids from diverse cancer types (including breast, colorectal, lung, neuroendocrine, ovarian, pancreatic, and prostate cancers). They utilized various biopsy sample types (such as core needle biopsies, paracentesis, or surgical samples) and considered different clinical settings (whether the patient had undergone chemotherapy and/or radiotherapy). Their results further supported the potential of tumor organoids in assessing treatment responses in patients [16,17]. Despite the promise of 3D models in the realm of research, their utilization comes with its own set of challenges. Firstly, the selection of a specific 3D culture method depends on the scientific inquiry at hand, with each method being tailored to address different research objectives [50,51]. However, several methodologies overlook the intricate interplay of the microenvironment in terms of its physical and chemical properties, especially concerning drug resistance. Additionally, employing static models, with or without scaffolds, might prove insufficient for studying dynamic processes such as metastasis, where fluid movements play a crucial role. Employing more intricate 3D models such as organoids or utilizing 3D bioprinting techniques could significantly enhance disease simulation and facilitate the development of personalized medicine programs. However, integrating complex 3D models into high-throughput screening processes may not be straightforward [50]. Furthermore, the heightened structural complexity of 3D cultures could pose challenges during analysis. One way to address this complexity is by integrating in silico models during analysis. The selection of the most appropriate 3D model hinges on the analytical processes that will be applied, and most existing analysis methods, which were initially designed for conventional 2D cell cultures, often struggle to adapt to 3D cultures and necessitate extensive validation steps [51]. Notably, 3D models encompass a more developed ECM, which can act as a barrier or a trap for chemicals and compounds. Unfortunately, this characteristic can be associated with issues related to diffusion during lysis or metabolic assays [50,51]. Nevertheless, these challenges should be faced during the development of novel models, which creates an opportunity for enhancement of the available technology in cell culture. Just as 2D culture methods prompted the development of analytical techniques, the increased use of 3D models will drive the innovation of new analytical approaches. The potential of 3D cultures lies in bridging the gap between in vitro and in vivo models. By augmenting the complexity of 3D models, while maintaining reliability and feasibility, in vitro researchers can come closer to the observations seen in vivo. Moreover, by incorporating computer modeling, it may be feasible to integrate 3D models into a more systemic environment.

The results from this scoping review will map the current utilization of these approaches in drug assessment in oral cancer. We aim to contribute to a better understanding of the suitability of three-dimensional models in preclinical drug trials and their potential to improve treatment development in oncology.

## Figures and Tables

**Figure 1 jpm-13-01618-f001:**
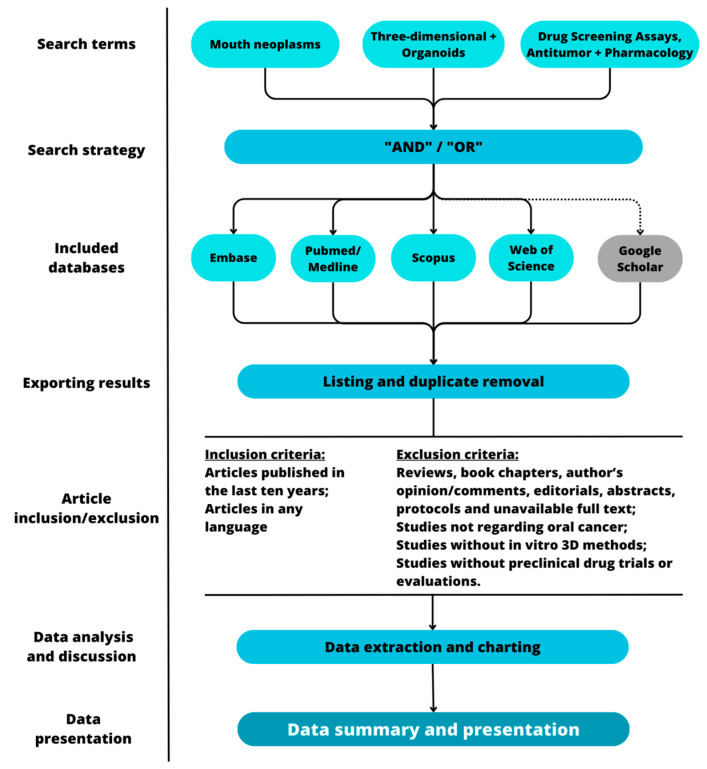
Graphic representation of the methodological steps of this study.

**Table 1 jpm-13-01618-t001:** The main population–concept–context mnemonic (PCC).

Population	Concept	Context
N/A	Three-dimensional models used for drug screening	Oral Cancer

**Table 2 jpm-13-01618-t002:** Search strategy.

Database	Strategy
PubMed/Medline	((“Mouth Neoplasms” [Mesh] OR “Mouth Neoplasms” OR “Mouth Neoplasm” OR “Neoplasm, Mouth” OR “Neoplasms, Oral” OR “Neoplasm, Oral” OR “Oral Neoplasm” OR “Oral Neoplasms” OR “Cancer of Mouth” OR “Mouth Cancers” OR “Oral Cancer” OR “Cancer, Oral” OR “Cancers, Oral” OR “Oral Cancers” OR “Cancer of the Mouth” OR “Mouth Cancer” OR “Cancer, Mouth” OR “Cancers, Mouth” OR “Oral Tongue Squamous Cell Carcinoma” OR “Oral Squamous Cell Carcinoma” OR “Oral Cavity Squamous Cell Carcinoma” OR “Oral Squamous Cell Carcinomas” OR “Squamous Cell Carcinoma of the Mouth”) AND (“Cell Culture Techniques, Three Dimensional” [Mesh] OR “Cell Culture Techniques, Three Dimensional” OR “3D Cell Culture” OR “3D Cell Cultures” OR “Cell Culture, 3D” OR “Cell Cultures, 3D” OR “Cultures, 3D Cell” OR “3-Dimensional Cell Culture” OR “3 Dimensional Cell Culture” OR “3-Dimensional Cell Cultures” OR “Three-Dimensional Cell Culture” OR “Cell Culture, Three-Dimensional” OR “Cell Cultures, Three-Dimensional” OR “Three Dimensional Cell Culture” OR “Three-Dimensional Cell Cultures” OR “3-D Cell Culture” OR “3 D Cell Culture” OR “3-D Cell Cultures” OR “Cell Culture, 3-D” OR “Scaffold Cell Culture Techniques” OR “Scaffold Cell Culture” OR “Cell Culture, Scaffold” OR “Cell Cultures, Scaffold” OR “Scaffold Cell Cultures” OR “Organoids” [Mesh] OR “Organoids” OR “Organoid”)) AND (“Drug Screening Assays, Antitumor” [Mesh] OR “Drug Screening Assays, Antitumor” OR “Antitumor Drug Screening Assays” OR “Cancer Drug Test” OR “Antitumor Drug Screen” OR “Anti-Cancer Drug Screens” OR “Anti Cancer Drug Screens” OR “Anti-Cancer Drug Screen” OR “Screen, Anti-Cancer Drug” OR “Anticancer Drug Sensitivity Tests” OR “Tumor-Specific Drug Screening Tests” OR “Tumor Specific Drug Screening Tests” OR “Pharmacology” [Mesh] OR “Pharmacology” OR “Pharmacologies”)

## Data Availability

The data presented in this study are available on request from the corresponding author.

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
