# Peer review of "Use of Three-Dimensional Cell Culture Models in Drug Assays for Anti-Cancer Agents in Oral Cancer: Protocol for a Scoping Review"

_jpm, 2023, doi:10.3390/jpm13111618_

Round 1
Reviewer 1 Report
Dear Authors,
thank you for submitting your protocol; my only suggestion would be to try to synthesize Table 1 thus being easier to analyze for the readers.
Reviewer 2 Report
This protocol lacks sufficient informative content and should be improved to better convey the study's purpose, methodology, and potential contributions. It is important to clearly outline the objectives, methods, and expected outcomes of the research. The protocol requires substantial improvement in terms of providing informative content. The authors briefly introduced the search protocol and keywords related to "Three-Dimensional Cell Culture Models in Drug Assays For Anti-Cancer Agents In Oral Cancer" but did not provide any details about the referenced papers or discuss their results. In contrast, researchers have been increasingly adopting patient-derived explant cultures to assess the effectiveness of anti-cancer agents across various cancer types. This personalized approach allows for the tailored evaluation of chemo drugs' efficacy, aiding in the selection of optimal treatments for chemotherapy. The authors did not include this widely used method of personalized drug screening, which has gained popularity, surpassing the utilization of 3D cultures. It is crucial for the manuscript to undergo significant improvements, incorporating relevant findings to bolster the review's scientific validity. The current presentation lacks the necessary rigor required by both the scientific community and patients.Author Response
Reviewer 2
1 - This protocol lacks sufficient informative content and should be improved to better convey the study's purpose, methodology, and potential contributions. It is important to clearly outline the objectives, methods, and expected outcomes of the research. The protocol requires substantial improvement in terms of providing informative content. The authors briefly introduced the search protocol and keywords related to "Three-Dimensional Cell Culture Models in Drug Assays For Anti-Cancer Agents In Oral Cancer" but did not provide any details about the referenced papers or discuss their results.
Response: Thank you for taking the time to review our protocol and for providing your feedback regarding its informative content. The submitted work is intended as a protocol paper detailing the methodology and data collection procedures without presenting results. Protocol work provides transparency to our planned approach and methodology, helps organize the process, identifies deviations from planned methods, helps to determine whether the bias impacts the interpretation of review results and conclusions, and helps to plan and outline the study methodology before embarking on the research. However, we recognize the importance of a full scoping review with results, data synthesis, and interpretation, and we plan to submit that as a separate manuscript once the study is completed.
We acknowledge the importance of clarity and ensure our audience effectively grasps the study's purpose, methodology, and potential contributions. To this end, we clearly outline a series of modifications to the study's purpose, methodology, and potential contributions. We hope modifications after this revision provide a comprehensive understanding of this protocol.
2 - In contrast, researchers have been increasingly adopting patient-derived explant cultures to assess the effectiveness of anti-cancer agents across various cancer types. This personalized approach allows for the tailored evaluation of chemo drugs' efficacy, aiding in the selection of optimal treatments for chemotherapy. The authors did not include this widely used method of personalized drug screening, which has gained popularity, surpassing the utilization of 3D cultures. It is crucial for the manuscript to undergo significant improvements, incorporating relevant findings to bolster the review's scientific validity. The current presentation lacks the necessary rigor required by both the scientific community and patients.
Response: We appreciate this comment on incorporating the significant role of patient-derived explant cultures in evaluating the effectiveness of anti-cancer agents across various cancer types. In light of your feedback, we affirm that data related to 3D patient-derived culture will be reached by question 1 (What subtype or subtypes of oral cancer were evaluated (e.g.: cell line, cell bank, origin, topographical location on the mouth) and question 3 (What was the three-dimensional model or models used (e.g.: spheroids, organoids, organ-on-a-chip) and their specifications (e.g.: protocol steps, use of scaffolds, patient-derived explants)?), as described in section 2.5 (data extraction and charting). In fact, we comprehensively have reviewed all of the items in this section, and the reformulated questions may be seen below:
- What subtype or subtypes of oral cancer were evaluated (e.g.: cell line, cell bank, origin, topographical location on the mouth)?
- What were the cell culture conditions (culture medium, additives, atmosphere, temperature)?
- What was the three-dimensional model or models used (e.g.: spheroids, organoids, organ-on-a-chip) and their specifications (e.g.: protocol steps, use of scaffolds, patient-derived explants)?
- What were the evaluated compounds and their relevant information (e.g.: purchase, concentration)? Were they antineoplastic agents?
- What was the treatment protocol (e.g: application scheme, duration, association with other treatments)?
- What were the control group(s)/conditions?
- What assays were performed to evaluate compound effects (e.g.: cell death, cytotoxicity, invasion, migration)? What were their results?
- What statistical analysis was performed? How many samples (e.g: number of spheroids or organoids) were used? How many assay repetitions were made?
- What are the authors' conclusions?
- Overall, does the study mention all data accurately?
Reviewer 3 Report
As the title indicates, in this paper, only the protocol is described. The manuscript lacks a results section; it solely details the methodology and data collection procedures. According to the Preferred Reporting Items for Systematic Reviews and Meta-Analyses Extension for Scoping Review (PRISMA-ScR) guidelines, the manuscript omits data synthesis and interpretation. While the conceptualization and data collection are comprehensively addressed, the sections on knowledge dissemination and application are inadequately articulated. Given these deficiencies, the manuscript may be rejected. Authors may be instructed to submit the article after completing the study.
Language is fine
Author Response
1 - As the title indicates, in this paper, only the protocol is described. The manuscript lacks a results section; it solely details the methodology and data collection procedures. According to the Preferred Reporting Items for Systematic Reviews and Meta-Analyses Extension for Scoping Review (PRISMA-ScR) guidelines, the manuscript omits data synthesis and interpretation. While the conceptualization and data collection are comprehensively addressed, the sections on knowledge dissemination and application are inadequately articulated. Given these deficiencies, the manuscript may be rejected. Authors may be instructed to submit the article after completing the study.
Response: Thank you for taking the time to provide insights on our manuscript. We respect the journal's rigorous standards and previously confirmed with the editorial office about acceptance of protocol-only submissions.
The submitted work is intended as a protocol paper detailing the methodology and data collection procedures preceding the presentation of results. Protocol work provides transparency to our planned approach and methodology, helps organize the process, identifies deviations from planned methods, helps to determine whether the bias impacts the interpretation of review results and conclusions, and helps to plan and outline the study methodology. Following guidelines from the Joanna Briggs Institute:
“An a priori protocol must be developed before undertaking the scoping review. A scoping review protocol is vital as it pre-defines the objectives, methods, and reporting strategies of the review, ensuring transparency in the process.” (Aromataris E, Munn Z (Editors). JBI Manual for Evidence Synthesis. JBI, 2020. Available from https://synthesismanual.jbi.global. https://doi.org/10.46658/JBIMES-20-01)
However, we recognize the importance of a full scoping review with results, data synthesis, and interpretation, and we plan to submit that as a separate manuscript once the study is completed. We recognize the significance of the PRISMA-ScR guidelines in ensuring quality and transparency. Given that our submission is intended as a protocol paper, it does not encompass data synthesis and interpretation. However, once completed, we assure you that the forthcoming scoping review will fully comply with the PRISMA-ScR guidelines, ensuring a comprehensive and rigorous presentation, as described in our methods section:
“2.6.2. Data summary
We intend to present our results in a narrative form, including tables containing the relevant extracted information. Findings will be described according to the review questions objectively and the results section may undergo further adjustments after the results are reviewed. As a guideline, the PRISMA-ScR checklist will be used in this review. Any further changes to the study protocol will be made as necessary and reported accordingly.”
We appreciate your attention regarding the sections on knowledge dissemination and application. Thank you again for your feedback. We remain committed to ensuring our work aligns with the high standards set by the journal and contributes valuable insights to the field. Please find a few examples of scoping review protocols previously published in scientific journals:
- Nkangu M, Obegu P, Asahngwa C, et al. Scoping review protocol to understand the conceptualisation, implementation and practices of health promotion within the context of primary healthcare in Africa. BMJ Open. 2021;11(12):e049084. Published 2021 Dec 2. doi:10.1136/bmjopen-2021-049084 (Availabe at: https://bmjopen.bmj.com/content/bmjopen/11/12/e049084.full.pdf)
- Dorfman TL, Archibald M, Haykowsky M, Scott SD. Correction: An examination of the psychosocial consequences experienced by children and adolescents living with congenital heart disease and their primary caregivers: a scoping review protocol. Syst Rev. 2023;12(1):98. Published 2023 Jun 20. doi:10.1186/s13643-023-02271-9 (Avaliable at: https://systematicreviewsjournal.biomedcentral.com/articles/10.1186/s13643-023-02271-9)
- van Staden D, Chetty V, Munsamy AJ. A protocol for a scoping review to map the assessment approaches in optometry education programmes globally. Syst Rev. 2022;11(1):33. Published 2022 Feb 21. doi:10.1186/s13643-022-01906-7 (Avaliable at: https://systematicreviewsjournal.biomedcentral.com/articles/10.1186/s13643-022-01906-7)
Reviewer 4 Report
The protocol for conducting the review titled "Utilization of Three-Dimensional Cell Culture Models in Drug Assays for Anti-Cancer Agents in Oral Cancer" appears to be well-constructed and methodologically sound. No significant methodological issues were identified during the assessment. Best of luck with your review process
Author Response
1 - The protocol for conducting the review titled "Utilization of Three-Dimensional Cell Culture Models in Drug Assays for Anti-Cancer Agents in Oral Cancer" appears to be well-constructed and methodologically sound. No significant methodological issues were identified during the assessment. Best of luck with your review process
Response: Thank you for your thorough assessment and positive feedback on our protocol study. We sincerely appreciate your recognition of the methodological soundness of our work.
Reviewer 5 Report
The article “Use Of Three-Dimensional Cell Culture Models In Drug Assays For Anti-Cancer Agents In Oral Cancer: Protocol For A Scoping Review” is very interesting and I have some comments to make to improve the article.
- Some questions are ambiguous or very general.
In the search strategy, no subtype or location of oral cancer appears
- What do you mean by subtypes of oral cancer?
- It would be very important to know the topographic area where the cancer is located, the types of drugs used for oral cancer, associated with other therapies or not...
- Why did you select the GRADE scale to assess the quality of the studies? Do you consider the five domains of the GRADE scale?
- The study has methodological limitations that can be resolved by analyzing the questions discussed in more detail.
Author Response
The article “Use Of Three-Dimensional Cell Culture Models In Drug Assays For Anti-Cancer Agents In Oral Cancer: Protocol For A Scoping Review” is very interesting and I have some comments to make to improve the article.
1 - Some questions are ambiguous or very general.
Response: A scoping review is designed to map the existing literature and evidence pertaining to a specific topic, necessitating a broader research question. This approach is distinct from a systematic review, which requires a more narrow, focused question aimed at comparing specific methods or techniques. Acknowledging this, we agree that our research questions could be formulated more directly and we thank you for this comment and opportunity to improve them. We have revised them accordingly to address this concern, as follows:
Original question: What 3D models are more frequently used in preclinical drug trials in oral cancer?
Revised question: Which 3D culture models have been predominantly utilized in preclinical drug trials for oral cancer treatment, and what are their primary applications?
Original question: How can three-dimensional culture methods be applied to preclinical trials of anti-cancer drugs in oral cancer?
Revised question: What protocols and techniques are recommended for applying three-dimensional culture methods in preclinical trials of anti-cancer drugs targeting oral cancer?
Original question: What advantages and disadvantages do 3D models bring to preclinical drug trials in oral cancer?
Revised question: What are the documented advantages and challenges of using 3D culture models in preclinical drug trials for oral cancer as compared to traditional 2D models?
2 - In the search strategy, no subtype or location of oral cancer appears.
Response: Thank you for the opportunity to make this point clear. By using terms like "mouth tumor" and "mouth neoplasm" we aim to cover the broad spectrum of oral cancers (e.g. squamous cell carcinomas, minor salivary gland carcinomas, adenocarcinomas etc.) that can appear in different locations within the mouth (e.g. buccal mucosa, oral mucosa, the floor of the mouth, hard and soft palates, tongue). However, we have added additional entry terms to the “Oral Cancer'' cluster, as described in the methods section:
“Mouth Neoplasms"[Mesh] OR "Mouth Neoplasms" OR "Mouth Neoplasm" OR "Neoplasm, Mouth" OR "Neoplasms, Oral" OR "Neoplasm, Oral" OR "Oral Neoplasm" OR "Oral Neoplasms" OR "Cancer of Mouth" OR "Mouth Cancers" OR "Oral Cancer" OR "Cancer, Oral" OR "Cancers, Oral" OR "Oral Cancers" OR "Cancer of the Mouth" OR "Mouth Cancer" OR "Cancer, Mouth" OR "Cancers, Mouth" OR "Oral Tongue Squamous Cell Carcinoma" OR "Oral Squamous Cell Carcinoma" OR "Oral Cavity Squamous Cell Carcinoma" OR "Oral Squamous Cell Carcinomas" OR "Squamous Cell Carcinoma of the Mouth"
3 - What do you mean by subtypes of oral cancer?
Response: We aim to cover the broad spectrum of oral cancers (e.g. squamous cell carcinomas, minor salivary gland carcinomas, adenocarcinomas etc.) that can appear in different locations within the mouth (e.g. buccal mucosa, oral mucosa, the floor of the mouth, hard and soft palates, tongue)
4 - It would be very important to know the topographic area where the cancer is located, the types of drugs used for oral cancer, associated with other therapies or not…
Response: We thank the reviewer for this relevant suggestion. With this in mind, we have revised our data extraction and charting questions (section 2.5) in order to include this information (as may be seen below):
- What subtype or subtypes of oral cancer were evaluated (e.g.: cell line, cell bank, origin, topographical location on the mouth)?
- What were the cell culture conditions (culture medium, additives, atmosphere, temperature)?
- What was the three-dimensional model or models used (e.g.: spheroids, organoids, organ-on-a-chip) and their specifications (e.g.: protocol steps, use of scaffolds, patient-derived explants)?
- What were the evaluated compounds and their relevant information (e.g.: purchase, concentration)? Were they antineoplastic agents?
- What was the treatment protocol (e.g: application scheme, duration, association with other treatments)?
- What were the control group(s)/conditions?
- What assays were performed to evaluate compound effects (e.g.: cell death, cytotoxicity, invasion, migration)? What were their results?
- What statistical analysis was performed? How many samples (e.g: number of spheroids or organoids) were used? How many assay repetitions were made?
- What are the authors' conclusions?
- Overall, does the study mention all data accurately?
5- Why did you select the GRADE scale to assess the quality of the studies? Do you consider the five domains of the GRADE scale?
Response: Assessing the quality of the studies included in scoping reviews is optional once the main objective is to identify knowledge gaps and clarify concepts. Given the lack of a standard quality assessment tool for in vitro studies, several authors tend to develop a checklist that suits the needs of their project (Tran et al., 2021). Authors agree that the GRADE scale was designed for systematic reviews instead of scoping reviews. However, we can benefit from the structure provided by GRADE to evaluate different components of the evidence once cancer three-dimensional cell culture models for drug discovery is a complex and multifaceted topic. Furthermore, GRADE will clarify evidence quality with a broad overview of the available evidence and also give an idea of where high-quality evidence exists (or is lacking), informing stakeholders broad strategic direction. In addition, our scoping review might serve as a precursor for a future systematic review, accelerating the transition between these two types of reviews. It is noteworthy that the GRADE scale has been a tool frequently used in an adapted form to analyze the quality of in vitro studies (e.g. Pavan et al., 2015, DOI: 10.1371/journal.pone.0130476; Magrin et al. - DOI : 10.3390/ijms21144895; de Morais et al. DOI: 10.1016/j.archoralbio.2020.104904). Furthermore, it is a tool that has been used to analyze the quality of studies in Scoping Reviews (e.g. Brainard et al., 2020, DOI: 10.2807/1560-7917.ES.2020.25.49.2000725; Dewhirst et al., 2022 - DOI: 10.1192/bjo.2022.4; Wooding et al., 2023 - DOI: 10.1097/ANA.0000000000000861). The five domains, risk of bias, inconsistency, indirectness, imprecision and publication bias, will be assessed.
Reference: Tran, L., Tam, D.N.H., Elshafay, A. et al. Quality assessment tools used in systematic reviews of in vitro studies: A systematic review. BMC Med Res Methodol 21, 101 (2021). https://doi.org/10.1186/s12874-021-01295-w
6 - The study has methodological limitations that can be resolved by analyzing the questions discussed in more detail.
Response: We sincerely appreciate your devoted time and effort to our manuscript. We have broadly revised this protocol according to your suggestions, providing a more comprehensive direction for readers. We hope the revised version addresses your concerns adequately, and that these modifications have enhanced the rigor and robustness of our protocol, bringing it closer in alignment with the high standards of JPM.
Round 2
Reviewer 5 Report
Thank you